# Comparison of Obturation Removal Efficiency from Straight Root Canals with ProTaper Gold or Reciproc Blue: A Micro-Computed Tomography Study

**DOI:** 10.3390/jcm9041164

**Published:** 2020-04-18

**Authors:** Vicente Faus-Matoses, Cristina Pasarín-Linares, Ignacio Faus-Matoses, Federico Foschi, Salvatore Sauro, Vicente J. Faus-Llácer

**Affiliations:** 1Department of Restorative Dentistry and Endodontics, Valencia University Medical and Dental School, 46010 Valencia, Spain; vicente.faus@uv.es (V.F.-M.); cpasarinlinares@gmail.com (C.P.-L.); ignacio.faus@uv.es (I.F.-M.); 2Department of Endodontics, Faculty of Dentistry, Oral & Craniofacial Sciences, Floor 22 Tower Wing, Guy’s Dental Hospital, London SE1 9RT, UK; federico.foschi@kcl.ac.uk; 3Institute of Dentistry, I. M. Sechenov First Moscow State Medical University, 119146 Moscow, Russia; salvatore.sauro@uchceu.es; 4Departamento de Odontologia, Facultad de Ciencias de la Salud, Universidad CEU Cardenal Herrera, 46115 Valencia, Spain

**Keywords:** micro-computed tomography, ProTaper Gold, Reciproc Blue, retreatment, Re-RCT, desobturation

## Abstract

Background: The objective was to evaluate the efficiency of ProTaper Gold (PTG) and Reciproc Blue (RB) NiTi files in obturation material removal from straight root canals assessed by micro-computed tomography. Methods: Fifty-two anterior human teeth were shaped with a PTG rotary system until F2 (25/.08). Specimens were obturated with a continuous wave of condensation technique. For retreatment, specimens were randomly distributed in two experimental groups: PTG group (F4 file) and RB (R40 file). Micro-CT scans were performed before and after retreatment procedures. The percentage of the remaining obturation material compared to the original volume was calculated, as well as the retreatment time. The presence of separated files was recorded. A descriptive analysis was carried out, and nonparametric tests were employed. Results: The mean percentages of remaining obturation material in the PTG group and the RB group were 4.14 ± 4.30% and 4.18 ± 4.29%, respectively. The mean retreatment times for the PTG and RB groups were 144 ± 51 and 163 ± 88 s, respectively. There were no statistically significant differences, neither in removal efficiency (*p* > 0.05) nor in retreatment time (*p* > 0.05), between the two groups. The coronal and middle thirds presented significantly more remaining obturation material than the apical third (*p* < 0.05). No file separation occurred. Conclusions: ProTaper Gold and Reciproc Blue present with comparable efficiency in removing the obturation material, with a similar mean retreatment time.

## 1. Introduction

The main treatment modality for teeth presenting with refractory infection in the presence of a failing root canal treatment is nonsurgical re-root canal treatment (Re-RCT) [1].

The achievement of apical patency in Re-RCT requires the removal of the pre-existing obturation material from the root canal space [2,3]. The most frequently used and taught obturation technique is gutta-percha in the form of cones in conjunction with a sealer [4,5].

The desobturation phase can be carried out with several approaches, including hand stainless steel instruments, rotary and reciprocating nickel–titanium (NiTi) instruments [6], specific NiTi retreatment files [6], ultrasonic, Gates Glidden burs and auxiliary methods, such as the use of solvents or heat [7].

In recent years, endodontic instruments have been improved in terms of design, kinematics and metallurgy [8]. Application of heat treatments to the NiTi alloy has increased flexibility and cyclic fatigue resistance [8,9,10].

The ProTaper Gold system (PTG, Dentsply Sirona, Ballaigues, Switzerland) and the single-file Reciproc Blue system (RB, VDW, Munich, Germany) are made from NiTi alloys that undergo complex heat treatments [11,12,13], maintaining the same geometric design of ProTaper Universal files (PTU, Dentsply Sirona) [8] and Reciproc (VDW) [12], respectively.

Heat treatment of nitinol provides tangible advantages in terms of increased resistance to separation and better flexibility [14]. The higher torsional fatigue of heat-treated alloy may be advantageous when dealing with pre-existing obturation material, thus reducing the chances of file separation during the desobturation phase [15]. A significant advantage of heat-treated NiTi files is to prevent the need for specific retreatment files to be added to the common endodontic armamentarium: retreatment-specific NiTi mechanical instruments have shown potential for iatrogenic damage due to cutting tips and stiffer structure [6].

A recent systematic review concluded that neither of the systems currently available are capable of completely removing obturation material from root-canal-filled teeth [6].

To date, there are no studies that evaluate the efficiency of heat-treated NiTi files in the removal of obturation materials from the root canal system. For this reason, the objective of this study was to compare the efficiency of PTG and RB files in the removal of gutta-percha from straight root canals using micro-computed tomography (micro-CT). The null hypothesis tested was that there were no statistically significant differences in the efficiency of these two systems.

## 2. Materials and Methods

### 2.1. Sample Size Calculation

Student’s *t*-test for independent variables, with an alpha-type error of 0.05, ±10 standard deviation, and 80% power to detect a determined effect size (*d* = 0.8), indicated that the minimum sample size required was 26 specimens per group (*n* = 52).

### 2.2. Specimen Selection

The study protocol was reviewed and approved by the Ethics Committee of Valencia University (Spain) (Review No: H1512486071276). Fifty-two extracted human incisors were utilized. Digital preoperative radiographs were taken in buccolingual and mesiodistal directions. Inclusion criteria were totally formed apices, the presence of a straight and oval root canal (according to the method of Schneider [16] and Jou et al. [17], respectively), and the presence of apical patency. The ratio of the buccolingual to the mesiodistal diameter of the canal was calculated at 5 mm from the apex.

All endodontic procedures were carried out by an expert endodontist, under magnification with an operating dental microscope (OPMI pico, Zeiss Dental Microscopes, Oberkochen, Germany).

### 2.3. Sample Preparation

Teeth were decoronated with a diamond disc (Brasseler USA, Savannah, GA, USA) to a standard root length of 16 mm. Apical patency was verified with a 10 K-file (Dentsply Sirona) until the tip was visible at the apical foramen under magnification. Working length was determined by subtracting 1 mm from this measure. Glide path presence was confirmed with a 15 K-file (Dentsply Maillefer, Baillagues, Switzerland).

### 2.4. Initial Root Canal Treatment

Specimens were shaped following the PTG sequence (Dentsply Sirona) up to F2 (25/08) filing with a torque-controlled X-Smart Plus motor (Dentsply Sirona), following manufacturer’s instructions. S1 and S2 files were used with brushing movements to working length. Subsequently, F1 and F2 files were taken to working length with a pecking motion. Between each file change, irrigation was carried out with 1 mL of 2.5% sodium hypochlorite (NaOCl) solution. Each file was employed in five root canals and then discarded. The final irrigation regimen consisted of 2 mL 2.5% NaOCl and 2 mL 17% EDTA for 2 min, followed by a final flush with 2 mL of 2.5% NaOCl. Root canals were dried with paper points (Dentsply Maillefer) and obturated with the continuous wave of condensation technique consisting of an apical plug and backfill using AH Plus as a sealer (Dentsply DeTrey, Konstanz, Germany). To verify root canal obturation quality, a radiograph was taken in the buccolingual direction. If voids were detected, the specimen was discarded and replaced. Access cavities were sealed with Cavit^TM^ (ESPE, Seefeld, Germany) and specimens were stored in a 100% humidity atmosphere at 37° for 14 days.

### 2.5. Retreatment Procedures

The 52 specimens were randomly distributed in two groups by a computerized algorithm (http://www.random.org): PTG group (Dentsply Sirona) and RB group (VDW).

No solvent was used during the retreatment procedures. Total retreatment time (in seconds) was recorded with a stopwatch, as well as the occurrence of any instrument fracture. Retreatment time did not include the time taken for irrigation and file changes. The desobturation phase was considered completed when no gutta-percha remnants were observed in the files’ flutes.

### 2.6. PTG Group

Following a similar protocol described previously in the literature (18), F4 (40/.06) and F3 (30/.09) files were used for the initial gutta-percha removal at the coronal and middle thirds, respectively. Then, the F2 (25/.08) file was taken to working length. Finally, the F3 and F4 files were employed until working length was reached. Each file was used in five root canals.

### 2.7. RB Group

Specimens were shaped with an R40 file (40/.06) (VDW) as recommended by the manufacturer with the “Reciproc All” program in the X-Smart Plus motor (Dentsply Sirona), with a sequence of three “in-and-out” motions combined with a lateral brushing action. After three pecking movements, the instrument was removed from the root canal and cleaned. The procedure was repeated until the file reached working length. Each file was used in one root canal.

After desobturation, the same final irrigation regimen as for primary root canal treatment was employed; finally, root canals were dried with paper points (Dentsply Maillefer).

### 2.8. Micro-Computed Tomography Scanning

Specimens were scanned with a micro-CT device (Micro-CAT II, Siemens PreClinical Solutions, Knoxville, Tennessee) with the following parameters: 80.0 kV, 500.0 uA, a 20 μm isotropic resolution and 360° rotation. Micro-CT images were automatically reconstructed by using the Cobra software (Exxim Computing Corporation, Pleasanton, CA, USA) and afterwards rendered by using the Amira 3D Software for preclinical analysis (Thermo Scientific, Waltham, MA, USA). The “watershed tool” was utilized to carry out the segmentation: two materials were selected, “root” and “root canal filling”; subsequently, all three axes (axial, coronal and sagittal planes) were automatically segmented, and, finally, the 3D reconstruction was displayed. Root volume was analyzed by using FIJI software (National Institutes of Health, Bethesda, MD, USA).

### 2.9. Statistical Analysis

Pre- and postoperative volumes of obturation material were expressed in mm^3^ and as a percentage of residual material from the original volume of the entire root canal. A descriptive analysis was performed for the percentage of remaining filling material and retreatment time. The hypothesis of normality and the equality of variances were evaluated using Kolmogorov–Smirnov and Levene tests, respectively. These tests normalized and equalized the variances in both groups. Medians were compared using a paired *t*-test. Finally, nonparametric tests were employed (Friedman, Wilcoxon, and Mann–Whitney tests) for comparation between thirds. The level of significance was set at *p* ˂ 0.05.

## 3. Results

The remaining obturation material expressed as percentage of the original volume was 4.14 ± 4.30% (CI: 2.41–5.88%) in the PTG group and 4.18 ± 4.29% (CI: 2.44–5.91%) in the RB group (Figure 1). The mean retreatment time was 144 ± 51 s (CI: 124–165 s) in the PTG group and 163 ± 88 s (CI: 127–199 s) in the RB group (Figure 2). No statistically significant differences were observed between the groups, neither in their removal of obturation material capacity (*p* > 0.05; *p* = 0.965) nor in the mean retreatment time (*p* > 0.05; *p* = 0.994) (Table 1).

Analysis of the three root thirds demonstrated that the coronal and middle thirds showed a comparable amount of remaining obturation material (*p >* 0.05; *p* = 0.702); conversely, a significantly lower amount of residual material was present at the apical third (*p* < 0.05; *p* = 0.004).

Ten specimens (six in the PTG group and four in the RB group) were completely free of remnants of obturation material after retreatment procedures. Micro-CT reconstructions of representative samples of both experimental groups are presented in Figure 3.

## 4. Discussion

The main cause of post-treatment apical periodontitis is the presence of a reoccurring or refractory intraradicular infection [18,19]. For this reason, complete removal of the pre-existing obturation material, followed by cleaning and shaping of the infected root canal walls, completed by a new obturation and an appropriate post-endodontic coronal restoration, remain the fundamental steps to achieve a successful outcome in nonsurgical retreatment [1,20,21]. An incomplete desobturation of the infected root canal space prevents a complete eradication of the intraradicular infection.

The primary outcome of the study was to determine the mean percentage of remaining obturation material within the root canal following the desobturation phase. There were no statistically significant differences between PTG and RB groups in their gutta-percha removal efficiency (*p* > 0.05). Currently, none of the available techniques is capable of completely removing gutta-percha and sealer from the root canal system [6,22,23,24], irrespective of the instruments employed. Due to the anatomic complexity of the root canal system [25] and the inability of endodontic instruments to reach all areas of the root canal [26], a complete removal of the obturation material is rarely achieved.

The majority of studies reported percentages of remaining obturation material in the root canal lower than 10% of the original volume [23,27], which is in accordance with the results of the present study.

The percentage of remaining gutta-percha in the coronal and middle third was significantly higher than in the apical third (*p* ˂ 0.05). This result differs from the findings reported by other authors [28]. In the present study, initial shaping of the root canal finished with the PTG F2 (25/.08) file. After retreatment, repreparation of specimens in both groups was concluded with the same apical diameter and taper (40/.06). Different studies reported that root canals should be reshaped to a larger apical size than in the initial root canal treatment [19,29]. These findings suggest that the diminished 0.06 taper of the files used at the retreatment stage may have a reduced efficieny at the coronal and mimddle third compared to the apical third, especially considering that the primary shaping was carried out with 0.08 taper files.

Procedural errors are untoward events that can occur during nonsurgical retreatment [30]. In the present study, no files separated. Although PTG and RB files were not specifically designed for retreatment cases, incorporation of thermal treatment protocols to the traditional NiTi alloy has allowed an increase in the cyclic fatigue resistance of endodontic instruments [8,9,10], which can be advantageous in the course of Re-RCT. Retreatment-specific files do not show particular advantages compared to heat-treated ones, which can, on the other hand, provide higher torsional resistance and flexibility [6]. In the present study, straight root canals were selected, therefore the resistance and efficiency in curved anatomy may have been not assessed.

Several methods have been employed for analysis of obturation material removal from root canals: for example, quantitative analysis of periapical radiographs [23], or longitudinal sectioning of the specimens and post-operative analysis with microscopes [31,32]; micro-CT is considered the gold standard as it makes it possible obtain 3D-high resolution images without destruction of specimens and introduction of significant artifacts [28,31,32,33].

The secondary outcome of the study was the mean retreatment time. There were no statistically significant differences between PTG and RB groups in the mean retreatment time (*p* > 0.05). This result is in accordance with the findings of Nevares et al. [28], who evaluated the efficacy of ProTaper Next and Reciproc files in obturation material removal. As in our study, the authors did not observe significant differences in mean retreatment time, taking into consideration the difference in the number of files. Previous studies also showed a quicker removal of gutta-percha utililsing re-treatement specific files; additionally, less remaining material was present, however this was based on rotary files [34]. On the other hand, there are studies that reported that the reciprocating technique is the most rapid method for gutta-percha removal [23]. Thus, the null hypothesis tested was accepted, as both experimental groups showed the same efficiency in removing obturation material in terms of time and residual remaining volume.

One of the limitations of the present study was that only one obturation technique was employed. In other studies, different techniques were compared, such as lateral compaction technique [23,29,33,35,36], single cone technique [22,37], Tagger’s hybrid technique [28,31] and continuous wave of condensation technique [10,27,38].

Another limitation of the study was the inclusion of straight root canals only as specimens. Curved root canals may offer more challenges during retreatment procedures [28,39], so it is possible that the results of the study cannot be extrapolated to those cases.

In addition, the results could vary depending on the operator’s experience, with a potential increase in the occurrence of procedural errors with less experienced operators.

## 5. Conclusions

PTG and RB files present similar capabilities in terms of obturation material removal and mean retreatment times. Comparable volumes of residual obturation material were present at the coronal and middle third of the canal. The apical third presented with a smaller volume of residual obturation material compared to the other two thirds of the canal.

## Figures and Tables

**Figure 1 jcm-09-01164-f001:**
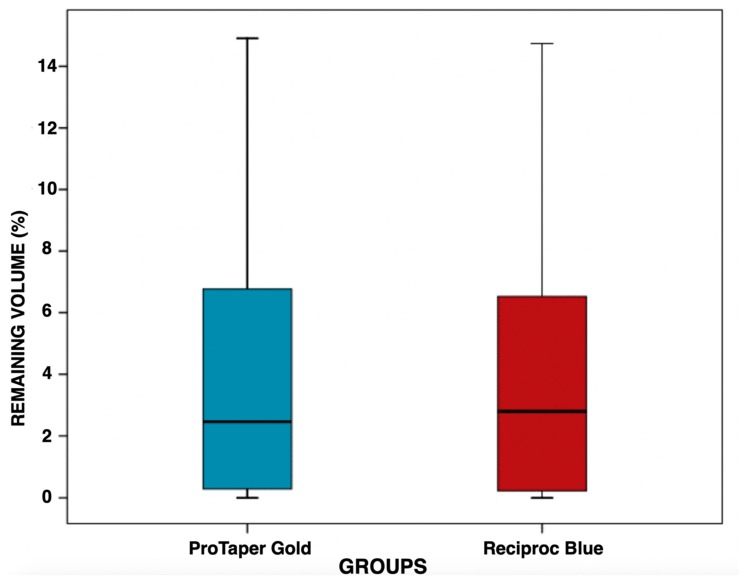
Boxplot showing the distribution of percentage of remaining obturation material in the ProTaper Gold and Reciproc Blue groups after retreatment. Note that the variability of the measures within each group is similar. The median is the horizontal line that divides the box.

**Figure 2 jcm-09-01164-f002:**
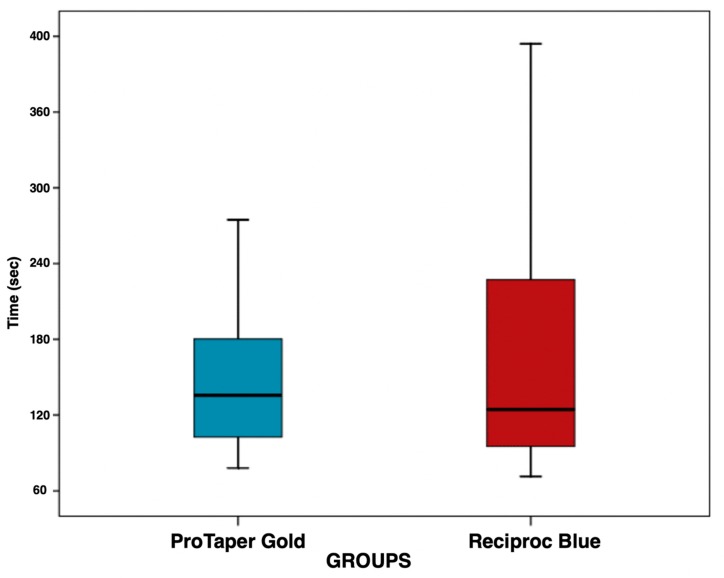
Boxplot showing the distribution of retreatment time in the ProTaper Gold group and Reciproc Blue group. The median is the horizontal line that divides the box.

**Figure 3 jcm-09-01164-f003:**
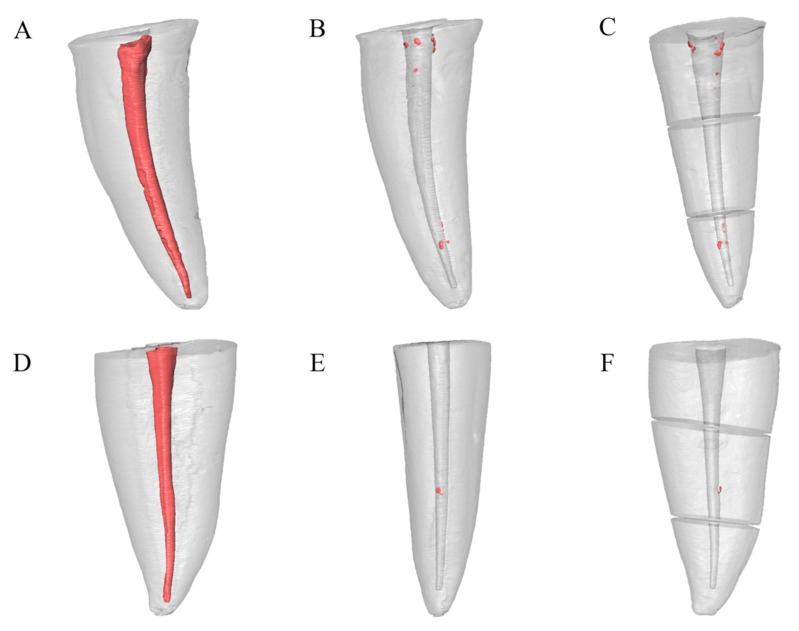
Micro-CT 3D reconstructions of representative samples of both experimental groups. (**A**–**C**) ProTaper Gold group. (**A**) Preoperative micro-CT image. (**B**) Postoperative micro-CT image. (**C**) Micro-CT image of root thirds. (**D**–**F**) Reciproc Blue group. (**D**) Preoperative micro-CT image. (**E**) Postoperative micro-CT image. (**F**) Micro-CT image of root thirds.

**Table 1 jcm-09-01164-t001:** Means and standard deviations of remaining obturation material (expressed as a percentage of the original volume) and means and standard deviations of retreatment time (expressed in seconds).

Groups	*n*	Mean Remaining Obturation Material (in Percentage) ± SD	Mean Retreatment Time (in Seconds) ± SD	*p* Value
**ProTaper Gold**	26	4.14 ± 4.30	147 ± 51	>0.05
**Reciproc Blue**	26	4.18 ± 4.29	163 ± 88	>0.05

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
