# Peer review of "Comparison of Obturation Removal Efficiency from Straight Root Canals with ProTaper Gold or Reciproc Blue: A Micro-Computed Tomography Study"

_jcm, 2020, doi:10.3390/jcm9041164_

Round 1

Reviewer 1 Report

The purpose of this study is to remove the root canal filling material, which is an essential technique in the treatment of re-infected root canals.

Author Response

Spelling checked as per suggestion

thank you

Reviewer 2 Report

The authors evaluated the obturation removal efficiency from straight canals with ProTaper Gold or Reciproc Blue. This topic is dealing with a clinically relevant issue and discusses the efficacy of two commercially available heat-treated Ni-Ti files in the removal of gutta-percha.

Introduction

Authors may describe further about the clinical implications of using heat treated files in re-root canal treatment in the introduction section. Potential advantages over specific Ni-Ti re-treatment files can be stated, which would be the clinical relevance of this study.

Materials and Methods

Page 3, Line 125

Please describe the specific protocols of measuring the volume of remained materials. The authors stated that Amira 3D software was used. In using Amira, segmentation criteria can affect the evaluation of binary-images based on grayscale levels.

Results

Page5, Line 163

One of the limitations of Micro-CT is that the radiopacity of filling materials may affect the results, although Micro-CT provides 3-D volumetric measurement with nondestructive approach. I suggest that the microscopic examinations of the sectioned sample could be helpful to confirm the results of the micro-CT, since the authors stated that the ten specimens were completely free of remnants of obturation material after retreatment procedure.

Discussion

Page7, Line 200

The authors may elaborate how an increased cyclic fatigue resistance can be advantageous in the course of re-RCT, and add relevant references.

Since the authors used the teeth with straight root canals, the risk of file separation might decrease.

Page7, Line 207

Please compare your results with those of other files specifically designed for retreatment cases. That would be the clinical implications of your study.

Author Response

Comments and Suggestions for Authors

The authors evaluated the obturation removal efficiency from straight canals with ProTaper Gold or Reciproc Blue. This topic is dealing with a clinically relevant issue and discusses the efficacy of two commercially available heat-treated Ni-Ti files in the removal of gutta-percha.

Introduction

Authors may describe further about the clinical implications of using heat treated files in re-root canal treatment in the introduction section. Potential advantages over specific Ni-Ti re-treatment files can be stated, which would be the clinical relevance of this study.

 The advantages of the heat treated nitinol are mentioned together with the potential advantages compared with specific re-treatment files.

Materials and Methods

Page 3, Line 125

Please describe the specific protocols of measuring the volume of remained materials. The authors stated that Amira 3D software was used. In using Amira, segmentation criteria can affect the evaluation of binary-images based on grayscale levels.

This is an important aspect and  the protocol utilised was explicated to clarify this aspect.

Results

Page5, Line 163

One of the limitations of Micro-CT is that the radiopacity of filling materials may affect the results, although Micro-CT provides 3-D volumetric measurement with nondestructive approach. I suggest that the microscopic examinations of the sectioned sample could be helpful to confirm the results of the micro-CT, since the authors stated that the ten specimens were completely free of remnants of obturation material after retreatment procedure.

 We agree that further inspection of the specimens with sectioning and optical microscope observation may have provided a further set of data however the wealth of research pertaining to micro-CT support the use of micro-CT as standalone imaging method.

Discussion

Page7, Line 200

The authors may elaborate how an increased cyclic fatigue resistance can be advantageous in the course of re-RCT, and add relevant references.

Since the authors used the teeth with straight root canals, the risk of file separation might decrease.

 These important considerations have been added to the discussion

Page7, Line 207

Please compare your results with those of other files specifically designed for retreatment cases. That would be the clinical implications of your study.

 Literature analysing specific retreatment files is cited

Reviewer 3 Report

·       Page 2, line 79: The authors state: “Fifty-two anterior human extracted teeth were utilized.” They should define what kind of anterior teeth were used in this study. Because of the great anatomic complexity of the root canal system, the maxillary canine has completely other dimensions than the mandibular central incisor. The dimensions of the specimens should be standardized.

·       In the initial root canal treatment, all specimens were shaped to a size 25/.08 file, which definitely is under-instrumentation for a maxillary canine for example.

·       Page 3, lines 98-100: The authors should define how the cervical and middle thirds of the canals were filled.

·       The authors should give an explanation for the Mean and SD in Fig 1. “The remaining obturation material expressed as percentage of the original volume was 4.14±4.30% (CI:2.41-5.88%) in PTG group and 4.18±4.29% (CI: 2.44-5.91%) in RB group (Fig 1)”. Could these numbers be due to variation in the dimensions of the samples?

·       The percentage of remaining gutta-percha in the coronal and middle third was significantly higher than in the apical third. This result differs from the findings reported by other authors and the authors should explain these differences.

Author Response

Comments and Suggestions for Authors

  • Page 2, line 79: The authors state: “Fifty-two anterior human extracted teeth were utilized.” They should define what kind of anterior teeth were used in this study. Because of the great anatomic complexity of the root canal system, the maxillary canine has completely other dimensions than the mandibular central incisor. The dimensions of the specimens should be standardized.
  • In the initial root canal treatment, all specimens were shaped to a size 25/.08 file, which definitely is under-instrumentation for a maxillary canine for example.

We specified that the teeth utilised were not including canine, otherwise the size would differ significantly and skew the results.

  • Page 3, lines 98-100: The authors should define how the cervical and middle thirds of the canals were filled.

As per methodology in the section initial root canal treatment a warm vertical condensation technique was used , now it is specified that it included apical plug and backfilling.

  • The authors should give an explanation for the Mean and SD in Fig 1. “The remaining obturation material expressed as percentage of the original volume was4.14±4.30% (CI:2.41-5.88%) in PTG group and 4.18±4.29% (CI: 2.44-5.91%) in RB group (Fig 1)”. Could these numbers be due to variation in the dimensions of the samples?

A certain variation was present as per SD however as per Kolmogorov test the distribution was normal p=0.032. Similarly the De Levene test confirmed an homogenous distribution.

  • The percentage of remaining gutta-percha in the coronal and middle third was significantly higher than in the apical third. This result differs from the findings reported by other authors and the authors should explain these differences.

Speculation re this finding has been provided.

Round 2

Reviewer 2 Report

Page4, Line 144

Authors stated that an automatic threshold segmentation approach was utilized.

Please describe the protocols more specifically. For example, the authors may explain how you select the region of interest, what kind of segmentation tools you used, and how you set the segmentation criteria. Amira software does not automatically calculate the volume, unless the observers set the measuring condition.  

Page5, Line 196

Please check the typos.

Author Response

Reviewer 2

Page4, Line 144

Authors stated that an automatic threshold segmentation approach was utilized.

Please describe the protocols more specifically. For example, the authors may explain how you select the region of interest, what kind of segmentation tools you used, and how you set the segmentation criteria. Amira software does not automatically calculate the volume, unless the observers set the measuring condition.  

 The tool used to carry out the segmentation was further specified

Page5, Line 196

Please check the typos.

Typos corrected, thank you

Reviewer 3 Report

The authors replied to the suggestions and improved the manuscript.

Author Response

Thank you